# PremPLI: a machine learning model for predicting the effects of missense mutations on protein-ligand interactions

Tingting Sun [1,2], Yuting Chen[1,2], Yuhao Wen[1], Zefeng Zhu [1] & Minghui Li [1✉]

Resistance to small-molecule drugs is the main cause of the failure of therapeutic drugs in clinical practice. Missense mutations altering the binding of ligands to proteins are one of the critical mechanisms that result in genetic disease and drug resistance. Computational methods have made a lot of progress for predicting binding affinity changes and identifying resistance mutations, but their prediction accuracy and speed are still not satisfied and need to be further improved. To address these issues, we introduce a structure-based machine learning method for quantitatively estimating the effects of single mutations on ligand binding affinity changes (named as PremPLI). A comprehensive comparison of the predictive performance of PremPLI with other available methods on two benchmark datasets confirms that our approach performs robustly and presents similar or even higher predictive accuracy than the approaches relying on first-principle statistical mechanics and mixed physics- and knowledge-based potentials while requires much less computational resources. PremPLI can be used for guiding the design of ligand-binding proteins, identifying and understanding disease driver mutations, and finding potential resistance mutations for different drugs. PremPLI is freely available at https://lilab.jysw.suda.edu.cn/research/PremPLI/ and allows to do large-scale mutational scanning.

[1] Center for Systems Biology, Department of Bioinformatics, School of Biology and Basic Medical Sciences, Soochow University, 215123 Suzhou, China. [2]These authors contributed equally: Tingting Sun, Yuting Chen. ✉email: minghui.li@suda.edu.cn

Protein–ligand interactions are of fundamental importance in myriad processes occurring in living organisms, triggering a multitude of signal transduction processes[1–3]. Many genetic diseases are caused by missense mutations through altering binding between small-molecule ligands and proteins[4,5]. For example, the association between disease-related mutations in the type I collagen and the ligand binding sites was observed[6] and the ligand-binding-domain mutations in human androgen receptor gene led to disrupted interaction between the N-terminal and C-terminal domains[7]. Moreover, drug resistance is the main cause of the failure of therapeutic drugs in clinical practice, particularly for the treatment of infectious diseases and cancer[8–10], and mutation in drug target is a critical mechanism that results in drug resistance[11–13]. For instance, in hormone-resistant breast cancer, mutations in estrogen receptor 1 gene (ESR1) are associated with acquired endocrine resistance[14], and two ESR1 ligand-binding site mutations produced partial resistance to the currently available endocrine therapies[15]. During the last decade, next-generation sequencing techniques have detected vast amounts of genetic mutations in humans, leaving clinicians and researchers without knowledge of whether these mutations are associated with genetic diseases or the emergence of drug resistance. Experimental methods can accurately measure the effects of missense mutations on proteins and their complexes, but they are time-consuming and expensive and do not have the capability to tackle large amounts of data[16]. Therefore, the development of reliable computational methods to reveal molecular effects of missense mutations would pave the way for the identification of pathogenic or drug resistance mutations and contribute to many biomedicine related fields and drug discovery.

Some attempts to model the effects of mutations on protein–ligand interactions have been made but with limited success and applicability. In summary, the methods can be classified into three main categories: (i) statistical/machine learning approaches using experimentally measured free energy data to parameterize regression models[17–19], such as mCSM-lig[17], which usually require little computational cost; (ii) simulation-based methods that rely on mixed physics-based and knowledge-based potentials to sample side-chain rotamers with restrained backbone motion, such as Rosetta[20], which have proven successful in the early stages of the discovery process[21,22]; (iii) alchemical free energy calculations that introduce a series of intermediate *alchemical* states in a thermodynamic cycle[23–27], which have become increasingly popular in the lead optimization stage of small molecule drug discovery and started being used prospectively by the pharmaceutical industry[28,29]. Several studies have been conducted to examine the performance of these methods on different test cases[19,25,27]. For instance, Aldeghi et al. evaluated the potential of all above three kinds of computational approaches to predict changes in binding affinity of eight different inhibitors with cancer target Abl kinase upon 144 mutations[19]. Overall, the prediction performance of Rosetta and alchemical free energy calculations are better than the current available machine learning approaches but they are still not satisfied and need to be further improved in both accuracy and speed. Alchemical calculations considering the full conformational flexibility of protein–ligand complex and the discrete nature of solvent are much more computationally expensive compared to statistical and (semi)-empirical approaches, moreover the workflow of which is complex, tedious, and error-prone. Therefore, developing prediction methods with tradeoffs between computational cost and accuracy is urgently required.

With a large amount of experimental data available, many data-driven machine learning methods have been developed to assess the impact of mutations on protein stability[30–36]

and protein–protein interactions[37–47], including PremPS[30], MutaBind[39], and MutaBind2[40] introduced by us. In addition, we also proposed PremPDI[48] and PremPRI[49] to evaluate the effects of mutations on binding between protein and nucleic acid. These approaches have achieved relatively high predictive accuracy with low computational cost. However, very few machine learning methods were proposed to assess the impact of mutations on protein–ligand binding and their prediction accuracy is very limited[17–19]. One reason hindering the development is due to the complexity of small molecule chemistry and binding interaction. Another major limitation is the availability of high-quality experimental data that can be used for training. Recently, a manually curated database Platinum was created[50], which associates experimentally measured effects of mutations on protein-small molecule binding with three-dimension structures of the corresponding complexes, allowing us to propose a structure-guided computational method to estimate the affinity changes upon mutations quantitatively.

Therefore, we developed a machine learning computational method, PremPLI, to estimate the effects of single point mutations on protein–ligand interactions by calculating the binding affinity changes quantitatively. PremPLI uses a random forest regression scoring function and consists of 11 sequence-based and structure-based features. It performs well across different types of cross-validation and independent tests, with similar predictive accuracy to Rosetta and alchemical free energy calculations but much lower computational costs. PremPLI can be used to aid in the development of new drugs to combat rising drug resistance, finding disease-causing or cancer driver missense mutations, and the design of proteins with novel ligand-binding functionalities and specificities.

## Results and discussion

Currently, the majority of studies for estimating the effects of mutations on protein–ligand binding are based on alchemical free energy calculations and Rosetta protocols. The prediction accuracy, especially the speed, are still not satisfied. Hence, we develop the machine-learning approach PremPLI (a flowchart highlighting the important steps in the methodology is provided in Fig. 1), which greatly improves the predictive speed without compromising the accuracy.

PremPLI predictive model is composed of 11 features and built using Breiman's random forest regression algorithm implemented in the R randomForest package. Hyperparameter tuning in a balanced 5-fold cross validation determined that the optimal settings for the number of decision trees and the number of features considered by each tree when splitting a node are 300 and 3, respectively. As described in the Methods section, we compiled two datasets of S859 and S796 (Fig. 2 and Supplementary Fig. 1 and Supplementary Table 1), and the difference between them is using all values or only one selected binding affinity change for these mutations with multiple $\Delta\Delta G_{exp}$. The performance trained and tested on S859 and S796 are shown in Supplementary Table 2 and no significant differences were observed. However, when tested on the blind set of S144 (Fig. 2 and Supplementary Fig. 2), the Pearson correlation coefficients using the models trained by S796 and S859 are 0.46 and 0.36, respectively, the difference is statistically significant (p-value < 0.01, Hittner et al.[51] test). Therefore, the unique single mutation dataset of S796 was used to parameterize PremPLI model. The Pearson correlation coefficient between experimental and calculated changes in binding affinity is 0.70 and the corresponding RMSE is 1.08 kcal mol$^{-1}$ when trained and tested on S796 (Fig. 3a). Besides, we also tried three other popular learning algorithms of support vector machine (SVM), eXtreme Gradient

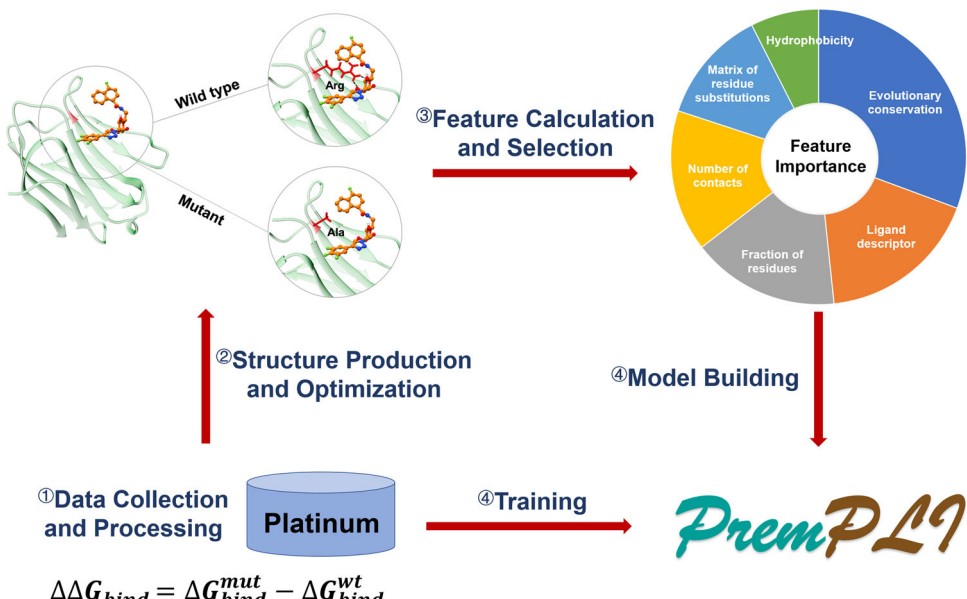

$$\Delta\Delta G_{bind} = \Delta G_{bind}^{mut} - \Delta G_{bind}^{wt}$$

**Fig. 1 A flowchart highlighting important steps in the methodology.** (1) Collecting and processing experimental data used for training, (2) Producing and optimizing 3D structures of wild-type and mutant protein-ligand complexes used for calculating structure-based features, (3) calculating around 400 features and selecting distinct features with remarkable contribution to the quality of the model, and (4) building PremPLI machine learning model using random forest algorithm and trained on experimental data.

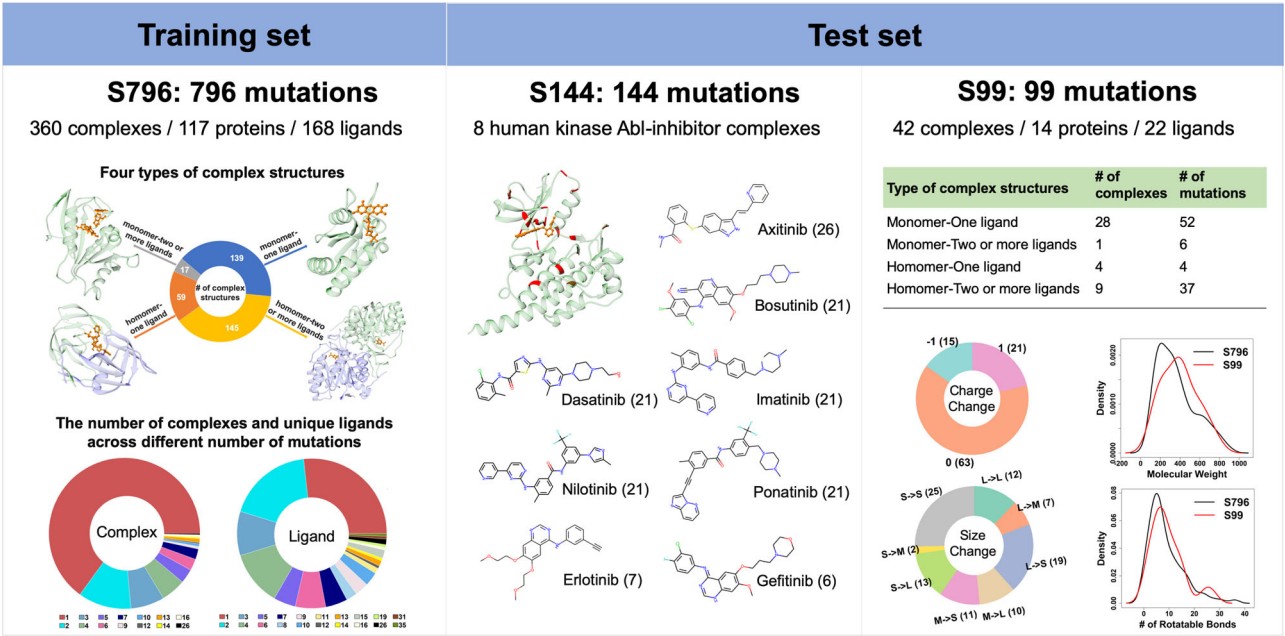

**Fig. 2 Overview of the data sets used.** S796: visualization of four types of protein–ligand complex structures and distribution of the number of complexes and unique ligands across different number of mutations are presented. Majority of complexes contain only one single mutation; S144: 3D structure of human Abl kinase with axitinib bound (PDB ID: 4WA9, mutation sites are shown in red), names and chemical structures of eight tyrosine kinase inhibitors (TKIs), and the number of mutations for each type of inhibitor are provided; S99: the number of complexes and mutations for each type of complex structure, statistics of the types of mutations (see Supplementary Fig. 1 for the definition), and distribution of molecular weight and number of rotatable bonds for the ligands in S796 and S99 are shown. See Supplementary Figs. 1 and 2 for more information about the data sets.

Boosting (XGBoost), and extremely randomized trees (Extra-Trees) to build the PremPLI model. Although the Leave-one-complex-out results provided in Supplementary Table 3 show that RF, XGBoost, and ExtraTrees are comparable in performance, the best performance is achieved by random forest when tested on the datasets of S144, S129, and S99 (Supplementary Table 4).

**Performance on four types of cross-validation.** In supervised machine learning, overfitting is a common problem in which the model performs well on the training data but poorly on the new and unseen data. To prevent overfitting, cross-validation has been used to tune the PremPLI model, and in addition, only 11 features with remarkable contributions were selected. Here, we further performed four different types of cross-validation to identify how

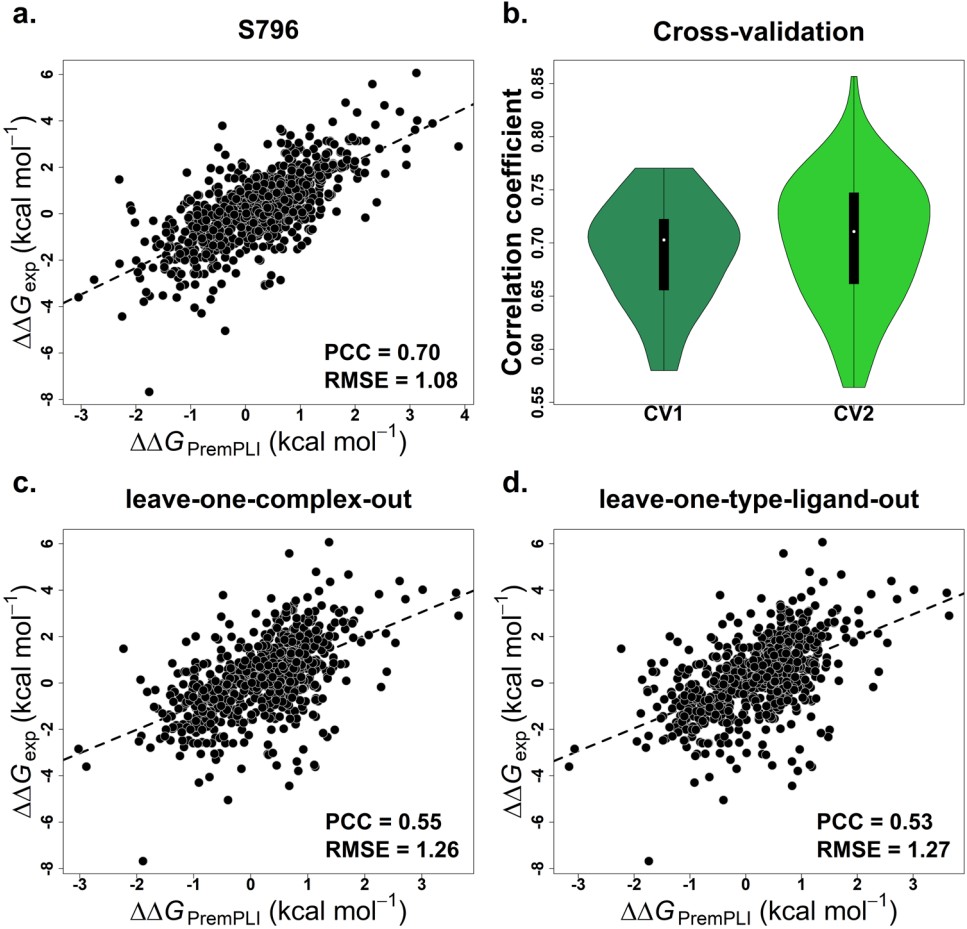

**Fig. 3 Pearson correlation coefficients between experimental and calculated changes in binding affinity. a** PremPLI trained and tested on S796 dataset, **b** ten times 5-fold and 10-fold cross-validations (CV1 and CV2), **c** leave-one-complex-out validation (CV3), and **d** leave-one-type-ligand-out validation (CV4). PCC Pearson correlation coefficient, RMSE (kcal mol$^{-1}$) root-mean-square error.

well our model performs on the unseen data (see "Methods" section for more details). Figure 3b shows the distribution of Pearson correlation coefficients for ten times 5-fold (CV1) and 10-fold (CV2) cross-validations. The PCC of each cross-validation round exceeds 0.56 with an average value of ~0.70 for both CV1 and CV2 (Supplementary Table 2). Upon the "leave-one-complex-out" procedure, the Pearson correlation coefficient between experimental and computed binding affinity changes reach the value of 0.55 and the corresponding RMSE is 1.26 kcal mol$^{-1}$ (Fig. 3c). Last, we performed a validation by leaving all complexes with the same type of ligand out of the training set ("leave-one-type-ligand-out" validation), and the correlation coefficient is 0.53, remaining statistically significant (Fig. 3d).

In addition, we analyzed 5% outliers to better understand strengths and limitations of PremPLI. Studentized residuals were used in detecting outliers. The performance after removing outliers is presented in Supplementary Fig. 3a, showing significantly improved PCC after removing 5% outliers. Through the analysis (Supplementary Fig. 3b, c), we found that the outliers are more likely to occur in complexes with lower-affinity ligand binding and at sites with higher number of proximal atoms and hydrogen bonds. Consistent with the observation from the study of ref. [17], outliers correspond to mutations with extreme experimental values (highly decreasing or increasing affinity). PremPLI could relatively correctly classify the highly decreasing mutations as decreasing but lose the ability to classify highly increasing mutations (Supplementary Fig. 3d).

**Validation on independent test sets and comparison with other methods.** First, the benchmark dataset of S144 (Fig. 2 and Supplementary Fig. 2) was used to assess the predictive performance of PremPLI and compare with other computational approaches. Hauser et al. calculated relative changes in free energy for these 144 mutations using MD simulations combined with the solution of the Generalized Born equation calculated by Prime and alchemical free-energy perturbation calculations using FEP+, respectively[25]. More recently, Aldeghi et al. also used this dataset and tested performance of different methodologies including MD simulations with a free energy calculation protocol, Rosetta, and machine learning (named as ML1 and ML2)[19]. For different prediction results of MD calculations under different force fields or Rosetta using different scoring functions, only the one showing the best performance was presented in our study. Table 1 provides the performance of six approaches tested on S144, which are PremPLI, FEP+ (MD calculation under the OPLS3 force field), R15 (Rosetta using the standard REF15 scoring function), Prime, and two machine learning methods of ML1 and mCSM-lig. The ML1 model trained on 484 single mutations from the Platinum database. mCSM-lig used a combination of different statistical potentials to predict $\Delta\Delta G$ values and was parameterized on 763 single mutations from Platinum database[17]. The uncertainties in the measures of PCC, RMSE, MCC, and the area under the ROC and PR curves evaluated by bootstrap are provided in Supplementary Table 5. The PCC and RMSE values shown in the Table 1, Supplementary Fig. 4a and Supplementary Table 5 indicate that R15 has the best performance, followed by PremPLI

**Table 1 Comparison of methods' performances on the datasets of S144, S129, and S99.**

| Dataset | Method | PCC | RMSE | AUC-ROC | AUC-PR | MCC |
|---|---|---|---|---|---|---|
| S144 | PremPLI | 0.46 | 0.77 | 0.78 | 0.36 | 0.40 |
|  | mCSM-lig[17] | 0.41 | 0.91 | 0.75 | 0.31 | 0.33 |
|  | ML1[19] | 0.12[a]** | 0.87 | 0.61* | 0.20 | 0.20 |
|  | R15[19] | 0.67** | 0.72 | 0.77 | 0.50 | 0.52 |
|  | Prime[25] | 0.29 | 1.81 | 0.67 | 0.27 | 0.35 |
|  | FEP+[25] | 0.49 | 1.07 | 0.76 | 0.53 | 0.52 |
| S129 | PremPLI | 0.48 | 0.78 | 0.81 | 0.35 | 0.39 |
|  | mCSM-lig[17] | 0.20[b]* | 1.06 | 0.55* | 0.28 | 0.40 |
| S99 | PremPLI[C] | 0.69 | 1.09 | 0.84 | 0.71 | 0.69 |
|  | A14[27] | 0.44** | 1.35 | 0.78 | 0.55 | 0.54 |
|  | R14[27] | 0.33** | 1.35 | 0.68 | 0.40 | 0.37 |
|  | RMD[27] | 0.48* | 1.23 | 0.77 | 0.59 | 0.58 |

* and ** indicate statistically significant difference between PremPLI and other methods in terms of PCC (Hitter et al.[51] test) and AUC-ROC (DeLong test) with $p$-value < 0.05 and $p$-value < 0.01, respectively. PremPLI[C]: PremPLI was retrained after removing all mutations in the overlapped complexes with S99 from the training dataset. R15: Rosetta using the *flex_ddg* protocol and REF2015 scoring function. R14: Rosetta using the *flex_ddg* protocol and talaris2014 scoring function. A14: Amber14sb and GAFF(v2.1)/AM1-BCC force fields were used for proteins and ligands, respectively. RMD: the combination of R14 and A14. The results of more combinations are shown in Supplementary Fig. 7.
All correlation coefficients are statistically significantly different from zero ($p$-value < 0.01, $t$-test) except [a]$p$-value = 0.14 and [b]$p$-value = 0.02.

and FEP+ methods, and ML1 performs the worst. In addition, Aldeghi et al. built a model of ML2 using ExtraTrees algorithm and training on the S144, and the PCC and RMSE are 0.57 and 0.68 kcal mol$^{-1}$, respectively, upon 8-fold nested cross-validation (similar to the leave-one-complex-out validation used in our study). To compare with ML2, we also used S144 as the training set and the ExtraTrees algorithm to build a new model. The PCC reaches 0.72 and RMSE is 0.60 kcal mol$^{-1}$ upon leave-one-complex-out validation, outperforming ML2 (Supplementary Table 6). Given that the two models used the same training set and algorithm, the main improvements come from the features being used.

Second, since most users using machine learning methods, such as PremPLI or mCSM-lig, will not or will not be able to process complex structures as Hauser et al. did for performing alchemical free-energy calculations, the workflow of which is relatively complex and tedious, we also calculated binding affinity changes of PremPLI and mCSM-lig using structures obtained from the Protein Data Bank directly (S129, 129 mutations from six Abl-TKI complexes). The results presented in Table 1 and Supplementary Table 5 demonstrate that PremPLI has significantly better performance than mCSM-lig and is not very sensitive to the initial input structures. Moreover, to further compare with mCSM-lig, we also retrained our model on S763 dataset (a training set of mCSM-lig, Supplementary Table 1). Even though our features selected were not based on this dataset, we still obtained a significantly higher correlation coefficient (PCC = 0.73 and RMSE = 1.07 kcal mol$^{-1}$) than mCSM-lig (PCC = 0.63 and RMSE = 2.06 kcal mol$^{-1}$)[17], both of which were upon 10-fold cross-validation.

Third, we assessed the predictive performance of PremPLI on S99 dataset. Since S99 is a subset of S796, we retrained the model after removing all mutations in the overlapped complexes with S99 from the training dataset, and then tested it on S99 (named as PremPLI[C], trained on 671 mutations from 318 complexes). Binding affinity changes for S99 calculated by 11 Rosetta protocols, first-principles statistical mechanics in which MD simulations used six different force fields, and their combinations were available from the study of

ref.[27]. The results of the best performing Rosetta protocol (R14) and MD calculation (A14), and their combination (RMD) were provided in Table 1 and Supplementary Table 5 and Supplementary Fig. 4b. The correlation coefficient of PremPLI[C] is 0.69, significantly higher than other methods ($p$-value < 0.01, Hitter et al.[51] test). On the S99, a diverse and challenging benchmark set, Rosetta did not perform as well as on the S144. In addition, we found that the prediction values of Rosetta for half of mutations from the S144 and S99 are around zero (Supplementary Fig. 4).

Last, it is important to put the performances of approaches in the context of their computational cost. The running time of PremPLI for a single mutation per protein with ~400 residues is about ten minutes on a single CPU core, and it requires ~20 s for each additional mutation introduced in the same complex. For instance, calculating $\Delta\Delta G_{PremPLI}$ for a mutation and 26 mutations in Abl-axitinib complex takes ~10 and 20 min, respectively. In addition, the computation time required for each feature is provided in Supplementary Table 7, and we could find that the calculation of PSSM takes up most of the stated time (9 min 30 s) since it performs PSI-BLAST[52] searches of protein sequences, but it does not need to be calculated again for each additional mutation. However, each $\Delta\Delta G$ estimate takes up to 32 h on a single CPU core and 72 h on a single GPU core for Rosetta (R15) and FEP+ calculations, respectively (Supplementary Table 8).

**Performance on predicting resistance mutations.** Generally, resistance can be defined according to the resistance fold (RF): RF ≤ 1, no resistance; 1 < RF ≤ 10, low resistance; RF > 10, resistance[19,25]. Following the equation of $\Delta\Delta G_{exp} = RT \ln RF$, no resistance, low resistance and resistance mutations correspond to $\Delta\Delta G_{exp} \leq 0$, $0 < \Delta\Delta G_{exp} \leq 1.36$ and $\Delta\Delta G_{exp} > 1.36$ kcal mol$^{-1}$, respectively. Here we examined the potential for PremPLI and other methods to predict resistance mutations. ROC and PR curves using different approaches to distinguish resistance from no and low resistance mutations are shown in Fig. 4. The values of area under the receiver operating characteristics curve (AUC-ROC) and precision-recall curve (AUC-PR) and maximal MCC values are provided in Table 1 and Supplementary Table 5, which are the most relevant performance measures to examine when the objective is to identify resistance mutations rather than assessing their impact on ligand binding quantitatively. The maximal MCC value was calculated for each method through calculating the MCC across a range of thresholds. Given the fraction of resistance mutations in the datasets of S144 and S99, a random classifier would return an AUC-PR of 0.13 and 0.19, respectively. In terms of the results provided in Table 1 and Fig. 4 and Supplementary Table 5, the best classifiers are PremPLI, Rosetta, and MD calculations.

As are shown in Supplementary Fig. 5, mutations located on binding interface have on average larger effects on protein–ligand interactions than non-interface mutations. The interface residues were defined if any heavy atoms of them are within 5 Å distance from any heavy atoms in ligands. Rosetta and MD calculations perform well on interface mutations but lose the ability to predict non-interface mutations, while PremPLI yields statistically significant positive correlations in predicting both interface and non-interface mutations (Table 2 and Supplementary Fig. 5). For instance, in S144 and S99, there are five non-interface resistance mutations, and the $\Delta\Delta G$ values of them are in the ranges of −0.06 to 0.13, −0.83 to 0.43, and 0.53 to 1.95 kcal mol$^{-1}$ predicted by Rosetta, MD calculations, and PremPLI, respectively.

**Performance on predicting different types of ligands.** Previous studies have demonstrated that the impacts of mutations on different types of ligands cannot be equally well predicted[19,25].

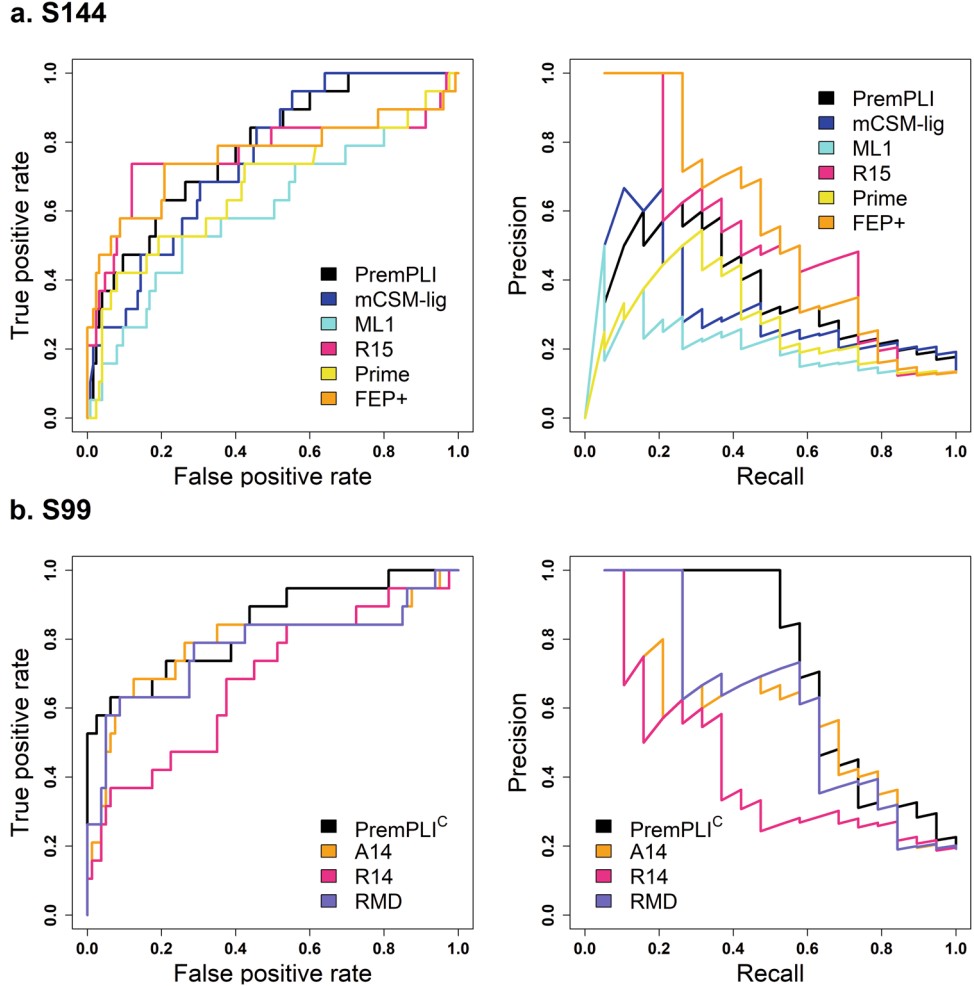

**Fig. 4 Receiver operating characteristics (ROC) and precision recall (PR) curves for different methods to distinguish resistance from other mutations.** The number of resistance mutations is 19 for both S144 (**a**) and S99 (**b**) datasets.

**Table 2 Pearson correlation coefficients between experimental and calculated ΔΔG values for interface and non-interface mutations.**

| Dataset | Method | Interface | | Non-interface | |
|---|---|---|---|---|---|
| | | **PCC** | **RMSE** | **PCC** | **RMSE** |
| S144 | PremPLI | 0.45 | 0.92 | 0.28 | 0.56 |
| | mCSM-lig[17] | 0.41 | 1.11 | — | 0.61 |
| | ML1[19] | — | 1.05 | — | 0.61 |
| | R15[19] | 0.74** | 0.82 | — | 0.59 |
| | Prime[25] | 0.25 | 2.40 | — | 0.76 |
| | FEP+[25] | 0.64* | 1.15 | −0.26** | 0.97 |
| S99 | PremPLI[C] | 0.70 | 1.22 | 0.67 | 0.75 |
| | A14[27] | 0.57 | 1.31 | — | 1.44 |
| | R14[27] | 0.36** | 1.48 | — | 1.02 |
| | RMD[27] | 0.59 | 1.25 | −0.36** | 1.19 |

Only correlation coefficients statistically significantly different from zero are shown ($p$-value < 0.05, $t$-test). * and ** indicate statistically significant difference between PremPLI and other methods in terms of PCC (Hitter et al.[51] test) with $p$-value < 0.05 and $p$-value < 0.01, respectively.

Here we further analyzed the predictive performance of PremPLI on each type of ligand. The S144 dataset is composed of eight TKIs, six of which have co-crystal structures and each has more than 20 mutations, and inhibitors of erlotinib and gefitinib adapted docking complex structures and each has less than ten mutations (Fig. 2 and Supplementary Fig. 2). Performance for each inhibitor calculated by six methods is shown in Fig. 5a and Supplementary Fig. 6. PremPLI and FEP+ perform well on four TIKs with statistically significant PCC values, and Rosetta produces statistically significant PCCs for six TKIs including two using docking models. All approaches lose the ability to predict the effects of mutations on ponatinib binding. By observing experimental and predicted values (Supplementary Fig. 6), we found that three mutations increasing Abl and axitinib binding are predicted by PremPLI as decreasing mutations. After excluding these three mutations, the correlation coefficients increase up to 0.43 for PremPLI ($p$-value < 0.05, $t$-test) but decrease for all other methods (Fig. 5b). Overall, Rosetta, FEP+, and PremPLI alternately present the best performance on different types of ligands, so it is necessary to develop new computational methods to complement each other.

In summary, Rosetta performs well on S144 dataset—with strong correlation, low absolute errors, and good classification performance, while it performs poorly on S99 dataset. MD calculations with a free energy calculation protocol perform robustly on different data with moderate prediction accuracy. However, for Rosetta or MD calculations, different scoring functions or force fields will present different prediction accuracies, and here we only chose and provided the results of the best performing protocols, so it would be an issue for applying

## a. S144

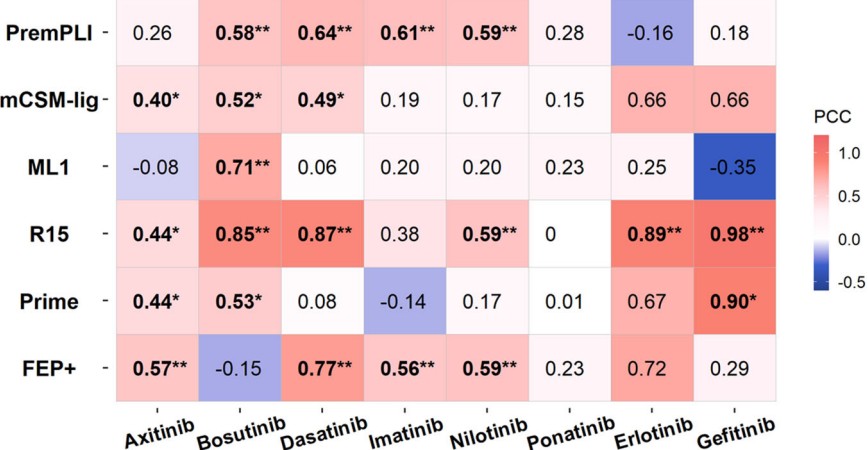

## b. Abl-axitinib complex

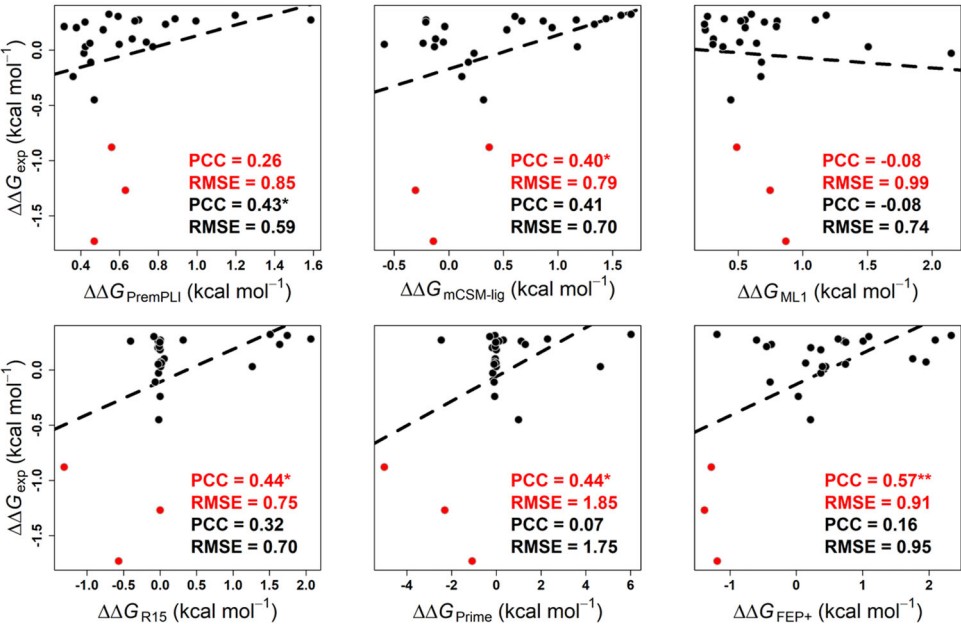

**Fig. 5 Performance for different methods tested on tyrosine kinase inhibitors. a** Pearson correlation coefficient for each tyrosine kinase inhibitor. * and ** indicate statistically significant difference from zero in terms of PCC with *p*-value < 0.05 and *p*-value < 0.01 (*t*-test), respectively. **b** Performance of six different methods tested on Abl-axitinib complex. PCC and RMSE in red: all 26 mutations; PCC and RMSE in black: mutations with three red dots removed.

them to unseen data. In addition, both approaches lose the ability to predict non-interface mutations and require heavy computational resources, especially for MD calculations. However, PremPLI presents similar predictive accuracy with the best performing Rosetta and MD calculation protocols while requiring much less computational resources. Therefore, a potentially integrated protocol might be PremPLI calculations for an initial large-scale mutational scan, followed by refinement of the most promising results via the combination of free energy calculations by Rosetta or MD simulations, which would allow for higher predictive power and provide dynamic insight into the impact of mutations.

Overall, in this work, we developed a machine learning approach for estimating protein–ligand binding affinity changes upon single mutations trained on a data set of 796 $\Delta\Delta G_{exp}$ values across 117 proteins and 168 ligands. As we know, three important aspects determine the prediction performance of a

machine learning method: the data used for training, the feature engineering, and the algorithm used to build the model. First, we handled the data obtained from the Platinum database very carefully (see Experimental datasets used for training PremPLI). Second, we calculated ~400 features and selected only 11 that have remarkable contributions to the quality of the model and can be explained from statistical and biological perspectives. Last, we tried four popular learning algorithms of random forest, Support Vector Machine, eXtreme Gradient Boosting, and Extremely Randomized Trees to build the PremPLI model, and selected the RF not only based on the performance on the training set but also on all three test sets. Through performing four types of cross-validation on the training set and the comprehensive validation on three independent diverse and challenging test sets (S144, S129, and S99), we believe the results presented provide a representative picture of the average performance for PremPLI.

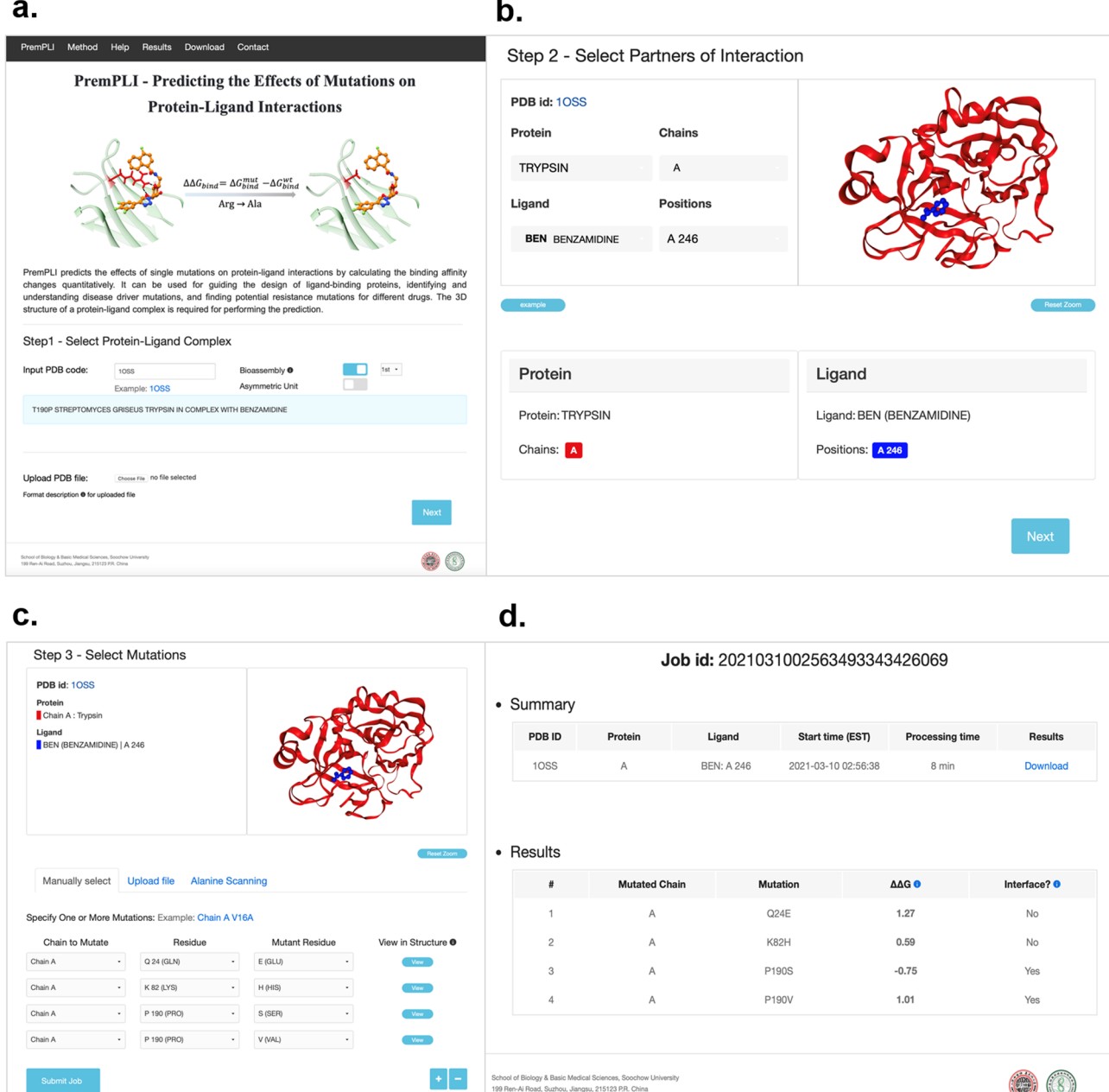

**Fig. 6 PremPLI server.** Three steps, (**a**) input Protein Data Bank (PDB) code or upload coordinate file, (**b**) select interaction partners and (**c**) assign mutations, and results pages (**d**) are provided. "Processing time" refers to the running time of a job without counting the waiting time in the queue.

**Online webserver**. The 3D structure of a protein–ligand complex is required by the webserver, and the user could input Protein Data Bank (PDB) code or upload the coordinate file in the standard PDB format. Biological assemblies or asymmetric unit can be chosen and retrieved from the Protein Data Bank when the user inputs the PDB code (Fig. 6). Next, the server will display a 3D view colored by protein chains and types of ligands and provide the corresponding protein and ligand names (Fig. 6). Given that PremPLI only calculates the impact of mutations on the interaction between a protein and one type of ligand, one or multiple chains that must belong to one protein and one or multiple ligands that must belong to one type can be assigned to the following calculation. The last step is to select mutations and three options are provided: "Specify One or More Mutations", in this option, users can not only perform calculations for the specified single mutations but also be allowed to view the mutated

residue in the complex 3D structure; "Upload Mutation List", allows users to upload a file including a list of mutations for performing large-scale mutational scans; "Alanine Scanning", allows users to perform alanine scanning for the selected protein chain (Fig. 6).

For each mutation, the PremPLI server provides $\Delta\Delta G$ (kcal mol$^{-1}$), predicted binding affinity change induced by this mutation (positive and negative sign corresponds to mutations decreasing and increasing affinity, respectively), and location of the mutation (interface/non-interface).

## Methods
**Experimental datasets used for training PremPLI**. Platinum[50], a manually curated database, includes experimentally measured effects of mutations (binding affinity changes) on structurally defined protein-ligand complexes. Binding affinity is calculated as $\Delta G = RT\ln(K_D) = RT\ln(K_i) = RT\ln(K_m)$, and the binding

affinity change upon mutation is defined as $\triangle\triangle G = \triangle G_{\text{mutant}} - \triangle G_{\text{wild-type}}$. The positive and negative values of $\triangle\triangle G$ correspond to the mutations decreasing and increasing binding affinity, respectively. A total of 797 single mutations with experimentally determined complex structures and binding affinity changes were included in the Platinum database, among which 151 mutations had both wild-type and mutant structures. Following the anti-symmetric property, that is, Gibbs free energy change introduced by a forward mutation (wildtype → mutant) plus the change induced by its reverse mutation (mutant → wildtype) should be equal to zero, so we took the mutant complex as being the "wild type" and inverted the sign of the affinity change, resulting in 948 mutations.

Next, we removed following entries from the Platinum database: (a) ligands or mutations cannot be found in the given complex structures; (b) complexes with modified residues at protein-ligand binding interface. If any heavy atom of a residue was located within 5 Å from any heavy atom of ligands, we defined this residue as an interface residue; (c) mutations with missing coordinates; (d) mutations occurring at metal coordination sites; (e) structures with only Cα atomic coordinates; (f) structures with a resolution lower than 3 Å. Besides, we only retained the interactions between one protein and one type of ligand. For mutations with multiple $\triangle\triangle G_{\text{exp}}$, we first preserved all values, resulting in 859 single mutations (it will be referred to as S859). Second, only one $\triangle\triangle G_{\text{exp}}$ value is chosen according to the following criteria in the order specified below: (1) pH is closest to neutral and/or the temperature (T) is closest to room temperature; (2) the value of $\triangle\triangle G_{\text{exp}}$ calculated by $K_D$ has the highest priority, followed by $K_i$; (3) the experimental method of fluorimetry is preferred over other measurements. As a result, 796 unique single mutations from 360 complex structures were retained (it will be referred to as S796). The majority of complexes contain only one mutation (Fig. 2). More information about the training set of S796 is shown in Supplementary Fig. 1 and Supplementary Table 1.

The last step is to process PDB coordinate files of complexes, which were obtained from the Protein Data Bank (PDB)[53]. First, the biological assembly was used for each complex. Second, only coordinates of proteins and ligands studied were retained. Third, the ligand was further removed from the PDB file when the ratio of the number of its contact heavy atoms to total heavy atoms is less than 0.5. We defined an atom in ligand as a contact atom when it is located within 6 Å from any heavy atom in protein. That is, only ligands that bind relatively strongly to protein were retained. As a result, four types of protein–ligand complex structures were included in the training set: monomer with one ligand; monomer with two or more ligands; homomer with one ligand; homomer with two or more ligands (Fig. 2). Please note, we only study the interactions between one protein and one type of ligand, but they may have multiple chains or multiple ligands.

**Experimental datasets used for testing.** Recently, Hauser et al. compiled a benchmark dataset consisting of reliable inhibitor $\triangle pIC_{50}$ data for 144 clinically identified mutants of human kinase Abl and examined the potential for alchemical free-energy calculations to predict resistance of these mutations to eight FDA-approved tyrosine kinase inhibitors (TKIs)[25]. None of the mutations or protein–ligand complexes in this benchmark dataset were included in our training dataset. The binding free energy change ($\triangle\triangle G_{\text{exp}}$) can be derived from the following equation:

$$\triangle\triangle G_{\text{exp}} = RT\ln\frac{K_{\text{i,mut}}}{K_{\text{i,WT}}} \approx RT\ln\frac{IC_{50,\text{mut}}}{IC_{50,\text{WT}}} \qquad (1)$$

Names, chemical structures, and the distibution of experimental $\triangle\triangle G_{\text{exp}}$ for each inhibitor are provided in Fig. 2 and Supplementary Fig. 2. In addition, Hauser et al. also provided eight processed co mplex structures (such as adding missing residues and loops) including two docking models, which were used to evaluate our method and perform comparisons with other methods (this dataset will be referred to as S144 including 144 mutations and eight processed complex structures). Since most users using our method will not or will not be able to produce the structures as Hauser et al. did for performing alchemical free-energy calculations in which the workflow is relatively complex and tedious, we also tested the performance of PremPLI on structures obtained from the Protein Data Bank directly (129 mutations from six Abl-TKI complexes, it will be referred to as S129). This can also check the sensitivity of our method to the initial input structures. The criteria for processing the training dataset have been applied in checking the test sets.

Another dataset proposed by Aldeghi et al.[27] includes 134 single and multiple mutations across 17 proteins and 27 ligands from the Platinum database. Only affinities determined by isothermal titration calorimetry and surface plasmon resonance were selected and proline mutations were excluded. Moreover, it is a diverse and challenging benchmark set, which includes large and flexible ligands, proteins with different folds, and many small-to-large/large-to-small and charge-changing mutations. Aldeghi et al. performed free energy calculations on these 134 mutations using first-principles statistical mechanics and Rosetta protocols, respectively. In our study, 25 multiple mutations were first removed from this benchmark. Then, ten single mutations for which ligand cannot be found in the given complex structures and protein interacts with multiple types of ligands were further removed. As a result, 99 single mutations from 42 complexes were retained (it will be referred to as S99, Fig. 2), which is a subset of S796.

The 3D structures of protein–ligand complexes were either taken from Hauser et al. study[25] (S144 dataset) or from the PDB (S796, S129, and S99 datasets). Mutant structures were produced using the BuildModel module of FoldX[34]. Missing heavy side-chain and hydrogen atoms in proteins were added via VMD program[54] using the topology parameters of CHARMM36 force field[55]. Hydrogen atoms of ligands were added via Chimera[56].

**The model of PremPLI.** PremPLI employs Random Forest (RF) regression scoring function. Around 400 features were calculated and considered in the model selection (Supplementary Table 9), and only 11 distinct features were found to contribute remarkably to the quality of the model (Supplementary Table 10).

- PSSM, a Position-Specific Scoring Matrix (PSSM) score for wild-type residue type at mutated site. PSSMs was calculated from PSI-BLAST[52] searches of protein sequences in NCBI non-redundant database and the resulting profile was constructed using the default parameters. $\Delta CS$ represents the change in conservation induced by a single mutation, which was calculated by PROVEAN program[57].
- $Prox^{\text{wt}}$ and $Prox^{\text{mut}}$ are the number of contacts between mutated site and ligands in wild-type and mutant structure, respectively. If the distance between two atoms is more than the Van der Waals interaction distance and within 5 angstroms, they were defined in contact. These terms were calculated by Arpeggio[58].
- $MWt$ presents the molecular weight of ligand which was calculated using XLOGP3[59]. $N_{\text{NO}}$ is the number of nitrogen and oxygen atoms of ligand.
- $H$ term accounts for the hydrophobicity of mutated site. The hydrophobicity for each type of amino acid residue was obtained from ref.[38].
- $P_{\text{RKDE}}$ is the fraction of charged residues (R, K, D, and E) in the wild-type structure. $P_{\text{RKDE}} = \frac{N_R + N_K + N_D + N_E}{N_{\text{All}}}$, for instance, $N_R$ is the number of all arginine residues, and $N_{\text{all}}$ is the total number of amino acids. $P_Q = \frac{N_Q}{N_{\text{All}}}$, $N_Q$ is the number of glutamine residues buried in the protein core. A residue is defined as buried if the ratio of solvent accessible surface area of this residue in the protein and in the extended tripeptide is less than 0.2[60].
- $M_{\text{AA1}}$ is the matrix of single residue interchanges derived from spatially conserved motifs[61] and $M_{\text{AA2}}$ is the matrix of amino acid substitutions produced by superposition of homologous protein structures[62]. They represent the more/less favorable trends of residue exchange in protein structures, obtained from Amino Acid Index Database with identifiers of AZAE970102 and RISJ880101 (AAindex, https://www.genome.jp/aaindex/).

**Cross-validation procedures.** We first performed ten times 5-fold and 10-fold cross-validations on the training dataset (named as CV1 and CV2, the folds were randomly split and clustered by mutations), and then evaluated the performance of our approach on two low redundancy sets; low redundant at (i) complex (named as leave-one-complex-out validation, CV3) and (ii) ligand (named as leave-one-type-ligand-out validation, CV4). There are 360 complexes and 168 types of ligands in the training dataset (Fig. 2). Namely, we leave all mutations from one complex (CV3) or from the complexes having the same type of ligand (CV4) out as a test set and use the remaining complexes/mutations to train the model, repeating this procedure for each complex or each type of ligand.

**Statistical analysis and evaluation of performance.** Pearson correlation coefficient (PCC) and root-mean-square error (RMSE) were used to measure the agreement between experimentally determined and predicted values of changes in binding affinity. Two-tailed $t$-test was used to assess whether the correlation coefficient is statistically significant from zero. Hittner et al.[51] and Fisher 1925[63] tests (two-sided) implemented in the R package cocor[64] were used to check whether the difference in correlation coefficients between PremPLI and other methods is significant. Hittner et al.[51] and Fisher 1925 tests are based on dependent groups with overlapping variable and independent groups respectively. RMSE (kcal mol⁻¹) is the standard deviation of the prediction errors, calculated by taking the square root of the average squared difference between predicted and experimental estimates of ΔΔG.

To quantify the performance of PremPLI and other approaches in distinguishing resistance from other mutations, we performed receiver operating characteristics (ROC) and precision-recall (PR) analyses. True positive rate/recall is defined as TPR = TP/(TP + FN), false positive rate is defined as FPR = FP/(FP + TN), and precision is defined as PPV = TP/(TP + FP) (TP: true positive; TN: true negative; FP: false positive; FN: false negative). In addition, we also calculated Matthews correlation coefficient (MCC) that accounts for imbalances in the labeled dataset.

$$\text{MCC} = \frac{\text{TP} * \text{TN} - \text{FP} * \text{FN}}{\sqrt{(\text{TP} + \text{FP})(\text{TP} + \text{FN})(\text{TN} + \text{FP})(\text{TN} + \text{FN})}} \qquad (2)$$

In addition, the uncertainties in these measures of PCC, RMSE, MCC, and the area under the ROC and PR curves were evaluated by bootstrap. Pairs of experimental and calculated ΔΔG were resampled with replacement 1000 times. From these 1000 bootstrap samples, 95% confidence interval was calculated and presented in the form of $x_{lower}^{upper}$, where $x$ is the mean statistic and the lower and upper bounds are the 2.5th and 97.5th percentiles. P-values for the differences

between PremPLI and other methods were also obtained from the bootstrap procedure. In each bootstrap sample, $\Delta\Delta G$ values from experiment, PremPLI and other methods to be compared were resampled together, and the difference in the performance metric of interest (e.g., $\Delta$PCC) between PremPLI and the approach to be compared was stored. A two-tailed $t$-test was used to test whether the difference (e.g., $\Delta$PCC) is statistically significant from zero.

**Reporting summary**. Further information on research design is available in the Nature Research Reporting Summary linked to this article.

## Data availability

Compiled experimental datasets and computational results that support our findings are publicly available on GitHub (https://github.com/minghuilab/PremPLI). Source data for graphs and charts are available at figshare[65].

## Code availability

Source code of PremPLI is available on GitHub (https://github.com/minghuilab/PremPLI)[66]. The PremPLI prediction model is available as a web-server tool at https://lilab.jysw.suda.edu.cn/research/PremPLI/.

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

## Acknowledgements
This work was supported by the National Natural Science Foundation of China [32070665] and the Priority Academic Program Development of Jiangsu Higher Education Institutions. The funders had no role in study design, data collection and analysis, decision to publish, or preparation of the manuscript.

## Author contributions
Conceptualization, M.L.; Methodology, T.S., Y.C., Z.Z. and M.L.; Software, T.S. and Y.C.; Validation, T.S. and M.L.; Formal analysis, T.S. and Y.C.; Investigation, T.S., Y.C. and M.L.; Data curation, T.S. and Y.C.; Writing—Original draft, M.L.; Writing—Review and editing, M.L.; Visualization, T.S. and Y.W.; Supervision, M.L.; Project administration, M.L.; Funding acquisition, M.L.

## Competing interests
The authors declare no competing interests.

## Additional information

**Peer review information**Communications Biology thanks the anonymous reviewers for their contribution to the peer review of this work. Primary Handling Editors: Debarka Sengupta, Luke Grinham, and Gene Chong. Peer reviewer reports are available.

