## [Transparent Peer Review File · Communications Biology]

Reviewers' comments:

Reviewer #1 (Remarks to the Author):

This article reports the development of PremPLI, a general machine learning model for the prediction of ligand binding affinity changes caused by protein mutation. I found the manuscript to be clear, fairly well written overall, and technically sound. The authors appear to objectively assess the performance of their ML model against established benchmarks and compare the results with other ML models as well as physics-based approaches like FEP+. PremPLI can be used by users without need of local installation via a web server, which also provides a summary of the methodology. In addition, the source code and datasets are openly available on GitHub. Because of the above reasons, I recommend the manuscript for publication. I have only the minor comments listed below that I would suggest the authors try addressing before publication.

- It seems the authors performed some form of feature selection in order to identify the final set of 11 features they report. It would be useful to know the details of how these features were selected; e.g. which other features were initially considered that were not selected for the final model? What was the rationale for creating the original pool of features?

- Performance metrics are reported without error estimates. I strongly encourage to report error estimates for all performance measures used, in particular when these measures are used to derive conclusions/observations about the performance of different methods. For instance, at page 10 it is stated that "Table S4 verified that the random forest algorithm shows the best performance". This is a bit unexpected since XGBoost is a fairly similar model and generally does not perform significantly worse than RF. Indeed, by looking at Table S4 one can see that the difference in performance between XGBoost and RF is so small (e.g., correlation of 0.55 vs 0.56) as to be insignificant. Standard errors of the mean on these performance measures are most certainly going to be larger than the difference between XGBoost and RF, suggesting the two approaches are in fact comparable in performance.

- How were the 5- and 10-fold cross-validation splits of the training dataset created? Were they random splits? Or were the folds clustered by, e.g., mutation, protein, ligand?

- At page 11, the authors compare PremPLI with the ML2 model of Aldeghi et al., noting how PremPLI performs better. Perhaps the authors could add one sentence speculating why this is the case. Given the two models are similar (both tree ensemble approaches), the main difference might come from the features being used?

- The computational cost of PremPLI is reported to be around 10 minutes per mutation. This is cheaper than simulation-based approaches, but still more costly than other ML models that can take seconds. I would imagine the cost is mostly due to the gathering of the required features, given the low cost of the RF algorithm itself. It would be interesting for the reader if the authors mentioned what aspects of the pipeline require most computing time, e.g. the calculation of which features takes up most of the stated 10 min time?

- At page 13-14: "MD calculations with a nonequilibrium free energy calculation protocol perform robustly on different data with moderate prediction accuracy". Is this a reference to the conclusions of another paper, just with a citation missing? Because in the discussed Figure 4 I could not find results about calculations with a non-equilibrium protocol. FEP+ uses a "traditional" equilibrium protocol.

- It would be beneficial for the reader to have an introductory figure and/or table that provides an overview of the datasets used. Table S1, Figures S2 and S3 go in this direction already, and they could be a starting point for an overview figure for the main text.

Reviewer #3 (Remarks to the Author):

In this paper, the authors proposed a structure-based machine learning method for quantitatively estimating the effects of single mutations on ligand binding affinity changes. The motivation and the experimental setup are clear. Overall, the paper is well organized.

Beside some minor issues described below, the main points of critics is that the technical description are not fully clear.

Thus, there is need to make these points more apparent.

1. As the proposed method contains several steps, to make the workflow more clear, it will be better if the authors could provide a flowchart to describe their method.
2. It will be better if the authors could provide more discussion about why is this particular method suitable for this particular computational problem (and type of data) from statistical and biological perspectives.
3. The proposed method starts with feature engineering, does this choice have a great influence on the results?
4. Besides those evaluation metrics and statistical analysis, it will be better if the authors could provide some case studies to demonstrate how does their method identify resistance mutations that cannot be detected by other methods.

We thank two reviewers for their time reviewing our paper and testing the web server; their comments and suggestions were very helpful and we believe we adequately addressed all of them.

Reviewer #1

This article reports the development of PremPLI, a general machine learning model for the prediction of ligand binding affinity changes caused by protein mutation. I found the manuscript to be clear, fairly well written overall, and technically sound. The authors appear to objectively assess the performance of their ML model against established benchmarks and compare the results with other ML models as well as physics-based approaches like FEP+. PremPLI can be used by users without need of local installation via a web server, which also provides a summary of the methodology. In addition, the source code and datasets are openly available on GitHub. Because of the above reasons, I recommend the manuscript for publication. I have only the minor comments listed below that I would suggest the authors try addressing before publication.

RESPONSE: We thank the reviewer for the positive comments.

1) It seems the authors performed some form of feature selection in order to identify the final set of 11 features they report. It would be useful to know the details of how these features were selected; e.g. which other features were initially considered that were not selected for the final model? What was the rationale for creating the original pool of features?

RESPONSE: We thank the reviewer for pointing this out. Yes, we performed feature selection to identify the final set of 11 features. Around 400 features that describe the characterization of protein, ligand, protein-ligand interaction, and mutation were calculated and considered. Only 11 features were selected that have significant contributions to the quality of the model and can be explained from statistical and biological perspectives. We added a table (Table S2) providing a brief description of all features initially considered in the model selection.

2) Performance metrics are reported without error estimates. I strongly encourage to report error estimates for all performance measures used, in particular when these measures are used to derive conclusions/observations about the performance of different methods. For instance, at page 10 it is stated that “Table S4 verified that the random forest algorithm shows the best performance”. This is a bit unexpected since XGBoost is a fairly similar model and generally does not perform significantly worse than RF. Indeed, by looking at Table S4 one can see that the difference in performance between XGBoost and RF is so small (e.g., correlation of 0.55 vs 0.56) as to be insignificant. Standard errors of the mean on these performance measures are most certainly going to be larger than the difference between XGBoost and RF, suggesting the two approaches are in fact comparable in performance.

RESPONSE: Per reviewer’s request, we added error estimates for all performance measures used. The uncertainties in the measures of PCC, RMSE, MCC, and the area under the ROC and PR curves were evaluated by bootstrap (see page 9 for more details). Four tables (Table S5-S8) were added to report the error estimates for all performance metrics used to derive observations about the performance of different methods tested on different datasets. As for the instance proposed by the reviewer, the previous results provided indeed cannot derive the conclusion that the random forest algorithm shows the best performance. This conclusion derived was also based on the performance of all algorithms tested on three independent data sets, so we added a table (Table S6) showing the performance on the test sets of S144, S129 and S99 using RF, SVM, XGBoost and ExtraTrees to build PremPLI model, respectively. We clarified this in the text.

3) How were the 5- and 10-fold cross-validation splits of the training dataset created? Were they random splits? Or were the folds clustered by, e.g., mutation, protein, ligand?

RESPONSE: The folds were randomly split and clustered by mutations in the 5- and 10-fold cross-validations. For leave-one-complex-out (CV3) and leave-one-type-ligand-out (CV4) validations, the folds were clustered by protein-ligand complexes and ligands, respectively. We clarified this in the text.

4) At page 11, the authors compare PremPLI with the ML2 model of Aldeghi et al., noting how PremPLI performs better. Perhaps the authors could add one sentence speculating why this is the case. Given the two models are similar (both tree ensemble approaches), the main difference might come from the features being used?

RESPONSE: We thank the reviewer for pointing this out. Aldeghi et al. built the ML2 model using ExtraTrees algorithm and training on S144. In the original version, we used S144 as the training set, the proposed 11 features and the RF algorithm to build a new model, outperforming ML2. In the revised version, we added XGBoost and ExtraTrees algorithms to build the model in addition to RF, and the Leave-one-complex-out results provided in Table S8 show that these three algorithms are comparable in performance. Then, we changed the PCC and RMSE values previously reported by RF to those calculated by ExtraTrees and added one sentence “*Given that the two models used the same training set and algorithm, the main improvements come from the features being used*” on page 12.

5) The computational cost of PremPLI is reported to be around 10 minutes per mutation. This is cheaper than simulation-based approaches, but still more costly than other ML models that can take seconds. I would imagine the cost is mostly due to the gathering of the required features, given the low cost of the RF algorithm itself. It would be interesting for the reader if the authors mentioned what aspects of the pipeline require most computing time, e.g. the calculation of which features takes up most of the stated 10 min time?

RESPONSE: We thank the reviewer for pointing this out. Yes, the cost is due to the gathering of the required features. The computation time required for each feature was added in Table S9, and we could find that the calculation of PSSM takes up most of the stated time (9min30s) since it performs PSI-BLAST searches of protein sequences, but it does not need to be calculated again for each addition mutation introduced in the same complex. We added description on page 13.

6) At page 13-14: “MD calculations with a nonequilibrium free energy calculation protocol perform robustly on different data with moderate prediction accuracy”. Is this a reference to the conclusions of another paper, just with a citation missing? Because in the discussed Figure 4 I

could not find results about calculations with a non-equilibrium protocol. FEP+ uses a “traditional” equilibrium protocol.

RESPONSE: We thank the reviewer for pointing this out. In the studies from Aldeghi et al. (refs#19 and #27), they reported the prediction results calculated by the approaches based on MD simulations with a nonequilibrium free energy calculation protocol. However, in this work, we only show the results tested on the S144 dataset obtained from the study of Hauser et al. (ref#25), and they used FEP+, a “traditional” equilibrium protocol. We have corrected it in the revised version.

7) It would be beneficial for the reader to have an introductory figure and/or table that provides an overview of the datasets used. Table S1, Figures S2 and S3 go in this direction already, and they could be a starting point for an overview figure for the main text.

RESPONSE: Per reviewer’s request, we added a figure (Figure 1) providing an overview of the datasets used.

Reviewer #3

In this paper, the authors proposed a structure-based machine learning method for quantitatively estimating the effects of single mutations on ligand binding affinity changes. The motivation and the experimental setup are clear. Overall, the paper is well organized.

Beside some minor issues described below, the main points of critics is that the technical description are not fully clear. Thus, there is need to make these points more apparent.

RESPONSE: We thank the reviewer for the positive comments.

1. As the proposed method contains several steps, to make the workflow more clear, it will be better if the authors could provide a flowchart to describe their method.

RESPONSE: We thank the reviewer for pointing this out. We added a flowchart (Fig. 2) highlighting the important steps in the methodology. In addition, we also added a figure (Fig. 1) providing an overview of the datasets used.

2. *It will be better if the authors could provide more discussion about why is this particular method suitable for this particular computational problem (and type of data) from statistical and biological perspectives.*

RESPONSE: Per reviewer's request, we added a paragraph on page 15 discussing why our computational method is suitable for this particular problem.

3. *The proposed method starts with feature engineering, does this choice have a great influence on the results?*

RESPONSE: Yes, feature engineering has a great influence on the results. We performed feature selection to identify the final set of 11 features. Around 400 features that describe the characterization of protein, ligand, protein-ligand interaction and mutation were calculated and considered, and only 11 were selected that have significant contributions to the quality of the model and can be explained from statistical and biological perspectives. We added a table (Table S2) providing a brief description of all features initially considered in the model selection. In addition, we compared PremPLI with the ML2 model of Aldeghi et al. and mCSM-lig using the same training set and algorithm, respectively, and PremPLI shows significantly better performance than ML2 and mCSM-lig (please see the first and second paragraphs on page 12 for more information). Therefore, we can conclude that feature engineering has a great influence on the results.

4. *Besides those evaluation metrics and statistical analysis, it will be better if the authors could provide some case studies to demonstrate how does their method identify resistance mutations that cannot be detected by other methods.*

RESPONSE: For predicting resistance mutations in S144 set, the best methods are R15 and FEP+, followed by PremPLI, while for S99, the best method is PremPLI, followed by MD calculations,

and Rosetta performs the worst. Through observing the experimental and predicted $\Delta\Delta G$ values, we found that Rosetta and MD calculations lose the ability to predict all resistance mutations (three in S144 and two in S99) that do not locate at the protein-ligand binding interface, while PremPLI is relatively successful in predicting them. The $\Delta\Delta G$ values of these five non-interface resistance mutations predicted by Rosetta, MD calculations and PremPLI are in the ranges of -0.06 to 0.13, -0.83 to 0.43, and 0.53 to 1.95 kcal mol⁻¹, respectively. We added description on page 14.

REVIEWERS' COMMENTS:

Reviewer #1 (Remarks to the Author):

The authors have satisfactorily address all my previous concerns and I am now happy to recommend the manuscript for publication.

Reviewer #3 (Remarks to the Author):

The authors have addressed all my concerns.